# Growth and Weight Status in Chinese Children and Their Association with Family Environments

**DOI:** 10.3390/children8050397

**Published:** 2021-05-14

**Authors:** Xu Tian, Hui Wang

**Affiliations:** 1College of Economics and Management, Academy of Global Food Economics and Policy, China Agricultural University, Beijing 100083, China; tianxu@cau.edu.cn; 2Department of Maternal and Child Health, School of Public Health, Peking University, Xueyuan Rd 38, Haidian District, Beijing 100191, China

**Keywords:** growth status, weight status, family environments, health differences

## Abstract

The growth status and weight status of Chinese children have experienced remarkable changes in the past decades. Using China Health and Nutrition Survey (CHNS) data, this paper examines the secular trends and disparity of the growth status and weight status in Chinese children and further investigates the impact of various family environments on children’s growth from 1991 to 2011. We found an increasing trend in standardized growth indicators (height, weight, and BMI), overweight, and obesity from 1991 to 2011. We also observed an increasing disparity in overweight and obesity over time. Family environments had a significant impact on children’s growth status and weight status. In particular, children that live in families with a small size, higher family income, better sanitary conditions, and with well-educated parents or overweight parents tended to be taller and heavier and have a higher BMI, lower risk of being underweight, and higher risk of exhibiting overweight and obesity. Further decomposition analysis showed that more than 70% of the disparity in standardized height, weight, and overweight and around 50% of the disparity in standardized BMI, underweight, and obesity could be attributed to heterogeneity in family environments. Moreover, the disparity associated with family environments tended to increase over time.

## 1. Introduction

An abundance of literature has shown that many developing countries have been undergoing a rapid nutrition transition, from a stage of receding famine to an increasing prevalence of diet-related noncommunicable diseases (DR-NCD) and obesity in the past few decades [1,2,3]. The improving nutrition sufficiency has resulted in a remarkable reduction in under-nutrition conditions, such as stunting and underweight in children, and a dramatically increased risk of overweight and obesity [3,4]. Between 2005 and 2016, the total number of people who were undernourished declined from 926 million to less than 800 million worldwide, and the prevalence of stunting in children aged under 5 declined from 29.5% to 22.9% [4].

China is one of the developing countries undergoing a rapid nutrition transition. Since 1980, Chinese consumers have been switching from traditional Chinese food, which is dominated by whole grains and vegetables, to diets high in fats, saturated fats, and sugar [5,6,7]. The changing diet, as well as declining physical activity and an increasing prevalence of a sedentary lifestyle [3,8], rapid economic growth and urbanization [1], cultural and technical change [9], and the decreasing number of children in each family [10], has contributed to a surging prevalence of overweight and obesity in China [11], particularly childhood obesity. Previous studies have shown that the prevalence of obesity has increased significantly since the 1980s for Chinese children in all age groups [12,13,14].

Moreover, the risk of overweight and obesity among children has been found to be associated with individual characteristics and family environments. For instance, the prevalence of obesity among boys was consistently higher than that for girls at all ages and across years [10,15,16], urban children and northern children were more likely to be obese [11,12,16], infants and children aged 10–12 had higher risks of overweight and obesity than children in other growth stages [11,16], and children with higher physical activity levels were associated with a significantly lower prevalence of overweight and obesity [3,10]. In addition, family background also played a significant role in shaping children’s body mass. The current literature suggests that children who grow up in more affluent families have a higher prevalence of overweight and obesity [17], and children with siblings are less likely to be overweight and obese than children in one-child families [10]. In addition, children’s risks of overweight and obesity are positively associated with parental education and BMI [10,17].

Both individual characteristics (e.g., age, sex, nationality, region, and intake of macronutrients) and social determinants such as family environments (e.g., parents’ characteristics, family income, number of siblings) play a big role in children’s growth and in shaping children’s weight status. However, individual characteristics are either given (e.g., gender) or under the control of children and can be changed via the children’s effort (e.g., dietary pattern, physical activity), while most family environments can be avoided by reasonable measures; the health differences associated with family environments are preventable and unnecessary and are usually considered health inequities, denoting an unjust difference in children’s health [18]. More importantly, ex-ante equality of opportunity is achieved if the environment does not affect children’s health outcomes. Therefore, the disparity of risks of overweight and obesity in children associated with family environments reflects social inequalities in health and can be reduced or eliminated. It is, therefore, particularly important to investigate the associations between family environments and children’s growth status and weight status and find appropriate measures to improve the overall health status of children.

To fill in the research gap in the current literature, this study extracted 20 years (1991–2011) of nutrition survey data from the China Health and Nutrition Survey (CHNS) (link: https://www.cpc.unc.edu/projects/china/data/datasets/index.html, accessed on 13 May 2021) and employed it to examine the secular trends and disparity of growth status and weight status in Chinese children. Moreover, the associations between various family environments and children’s growth status and weight status were also investigated.

## 2. Materials and Methods

### 2.1. Study Design and Participants

Participants in this study were selected from eight waves (1991, 1993, 1997, 2000, 2004, 2006, 2009, and 2011) of the China Health and Nutrition Survey (CHNS). The CHNS is an ongoing, nationally representative survey that is jointly conducted by the Carolina Population Center at the University of North Carolina at Chapel Hill (NC, USA) and the National Institute for Nutrition and Health at the Chinese Center for Disease Control and Prevention (CCDC) (Beijing, China).In each wave, about 4400 households were selected through a multistage, random cluster survey in nine provinces of mainland China (Liaoning, Heilongjiang, Jiangsu, Shandong, Henan, Hubei, Hunan, Guangxi, and Guizhou; three megacities, Beijing, Chongqing, and Shanghai, only joined the survey in 2011). More detailed information about the survey design and methods has been described elsewhere [19]. The CHNS collects comprehensive information on anthropometric measures, socioeconomic characteristics, and food consumption for individuals and families. Participants with incomplete physical examination data were deleted (*n* = 96,576). In particular, we only focused on children aged between 0 and 18 years old (*n* = 21,307). Children with biologically implausible values (a standardized BMI greater than 3 or less than 0.5) were excluded (*n* = 15). Finally, we collected data for 21,292 children with complete growth measures. Furthermore, we combined children’s physical examination data with children’s characteristics (age, sex, ethnicity, region, and calorie sources), family background (number of children, household size, per capita income, sanitation conditions, and parental education attainments), and parents’ hereditary factors (parents’ weight, height, and BMI) to investigate the determinants of children’s growth status and weight status. A certain number of participants was lost due to incomplete information (*n* = 6033), and the final matched sample size used in the association analysis between family characteristics and children’s health outcomes was 15,259, which was roughly equally distributed in the eight waves. The flow chart in Appendix A provides information on the selection of participants.

### 2.2. Measurements of Variables

The weight and height of each individual in the CHNS were measured by trained health workers using regularly calibrated equipment and according to the manufacturer’s instructions (SECA880 scales and SECA 206 wall-mounted metal tapes). BMI (kg/m^2^) was calculated as body weight (kg) divided by height squared (m^2^). In order to remove the variation in children’s growth status associated with age and gender, we normalized children’s height, weight, and BMI using the WHO 2007 growth reference [20,21]. We further adopted the WHO cut-offs to classify children in this study into groups of underweight (BMI less than 1 standard deviation below the median), normal, overweight (BMI more than 1 standard deviation above the median), and obesity (BMI more than 2 standard deviation above the median) [22]. One aspect that we need to mention is that the WHO growth reference is slightly different from the Chinese guideline (see Appendix A) [23,24,25,26]. However, the results were not very different when the Chinese reference was employed, and we therefore used the WHO standard here for comparison across different countries. Other individual characteristics (age, gender, ethnicity, and residential region) were measured by dummy variables. The dietary patterns of children were captured by the share of calorie sources drawn from fat (fat_share) and protein (protein_share). The secular trend of health outcomes is illustrated in two ways: first, using a figure to show the fluctuation of value over time; second, using a linear trend test to investigate whether the health outcome changed linearly over time.

Parents’ height, weight, and BMI were calculated using the same method. In addition, we further classified families into four groups based on the parents’ BMI: Group 1 (none of the parents’ BMI was ≥24, and at least one of them had a BMI of <18.5), Group 2 (one parent’s BMI was smaller than 18.5, but the other parent’s BMI was between 24 and 28, or both parents’ BMI were between 18.5 and 24), Group 3 (both parents’ BMI scores were greater than or equal to 18.5 but smaller than 28, and at least one of them had a BMI greater than or equal to 24 but smaller than 28), and Group 4 (at least one of parents’ BMI was ≥28).

The household scale was measured by two variables: household size (number of household members living in the family) and number of children (number of children who lived in the same household, including stepchildren). Households were further classified into three groups: small household (only one child), middle household (more than one child but no more than five members), and large household (more than one child and more than five members). Per capita household income was measured by the generated per capita gross income, which is the sum of all sources of income and revenue (business, farming, fishing, gardening, livestock, nonretirement wages, retirement income, subsidies, and other income). The consumer price index (CPI) was used to deflate all income into the 1991 constant price. Finally, real per capita household income (after deflation) was used to classify all families into three equal fractals (low, middle, and high-income groups). The sanitation condition of a household was measured by three variables: drinking water (in-house or in-yard tap water), toilet (in-house flush or in-house toilet), and basic sanitation (no near-house excreta removal). Families were classified into four groups based on the summary of these three sanitation variables (0 to 3). The education of parents was measured by the formal education year acquired. Furthermore, we classified parents’ education attainments into three groups: low-education family (none of the parents received more than 6 years of formal education), middle-education family (at least one parent received 6–12 years of formal education, but none received more than 12 years of education), and high-education family (at least one of the parents received more than 12 years of education).

### 2.3. Empirical Model

Multivariable linear regressions were employed to investigate the associations between growth status and family backgrounds after adjusting for individual heterogeneity. As the data were strongly unbalanced, pooled ordinary least square estimation (OLS) and two static panel models—Random Effect (RE) and Fixed Effect (FE)—were adopted in the estimation. The preferred model was chosen according to three model specification tests: the F test (FE vs. OLS), the LM test (RE vs. OLS), and the Hausman test (RE vs. FE).
(1)$$y_{it}=\beta_{0}+X_{it}\beta +Z_{it}\delta +v_{i}+u_{it}$$

Here, $y_{it}$ refers to three growth indicators (standardized height, weight, and BMI), $X_{it}$ denotes individual heterogeneity (age, gender, ethnicity, calorie sources, residence location), and $Z_{it}$ refers to various family environments, including two household scale variables (household size and number of children), family income per capita, three variables to measure sanitary conditions (access to tap water, ownership of private toilet, and availability of basic sanitation service to remove near-house excreta), maximum education, and the BMI of parents.

Two logistic models (pooled logistic regressions and RE logistic regression) were used to assess the association between weight status (underweight, overweight, and obesity) and family backgrounds after adjusting for individual characteristics. The preferred model depended on whether the panel-level variance component was important (the Likelihood Ratio test).
(2)$$\ln\frac{P(h_{it}=1|X_{it},Z_{it})}{1-P(h_{it}=1|X_{it},Z_{it})}=\beta_{0}+X_{it}\beta +Z_{it}\delta +v_{i}+u_{it}$$

Here, $h_{it}$ is a binary variable measuring the three weight status variables of children (underweight, overweight, and obesity).

### 2.4. Measurement and Decomposition of Disparity

Mean log deviation (*GE*(0)) was adopted to measure the disparity of continuous indicators because it is additively decomposable by population subgroup [27].
(3)$$GE(0)=\frac{1}{N}\sum_{i}^{N}\ln\left(\frac{\bar{x}}{x_{i}}\right)$$

Here, $x_{i}$ represents various growth indicators, and $\bar{x}$ is the mean value of $x_{i}$.

The disparity in the three weight status variables was captured using the modified dissimilarity index (*DI*).
(4)$$DI=\frac{2}{N}\sum_{i}^{N}\left|x_{i}-\bar{x}\right|$$

Further, total disparity was decomposed into within-group disparity and between-group disparity for various groups, defined by family backgrounds, including three household scale groups, three income groups, four sanitation groups, three parents’ education groups, and four parents’ BMI groups.
(5)$$\begin{array}{l} GE{\left(0\right)}_{b}=\sum_{j=1}^{J}\frac{N_{j}}{N}\ln\frac{\bar{x}}{\bar{x_{j}}}\\ GE{\left(0\right)}_{w}=\frac{1}{N}\sum_{j=1}^{J}\sum_{i=1}^{N_{j}}\frac{\bar{x_{j}}}{x_{i}} \end{array}$$

Here, $GE{\left(0\right)}_{b}$ is the between-group disparity and $GE{\left(0\right)}_{w}$ is the within-group disparity, while $\bar{x_{j}}$ and $N_{j}$ are the mean value and number of observations in group j.
(6)$$\begin{array}{l} DI_{b}=\frac{2}{N}\sum_{j=1}^{J}N_{j}\left|\bar{x_{j}}-\bar{x}\right|\\ DI_{w}=\frac{2}{N}\left(\sum_{i}^{N}\left|x_{i}-\bar{x}\right|-\sum_{j=1}^{J}N_{j}\left|\bar{x_{j}}-\bar{x}\right|\right) \end{array}$$

Similarly, $DI_{b}$ and $DI_{w}$ refer to between-group and within-group disparity, respectively.

To further calculate the part of disparity associated with family environments, we adopted the method proposed by Ferreira–Gignoux [28] and Soloaga and Wendelspiess [29] to estimate the ex-ante disparity associated with family background (inequality of opportunity (*IOP*)) for the continuous variable (physical growth) and binary variable (weight status), respectively.
(7)$$IOP=\frac{I(\overset{\wedge }{x})}{I(x)}$$

Here, $\overset{\wedge }{x}=E(x|C)$ is the expected health outcome under the conditions of various family environments. I(x) and $I(\overset{\wedge }{x})$ are the disparity indicators (the aforementioned *GE*(0) and dissimilarity index) calculated using actual and fitted values of health outcomes. Therefore, Equation (7) gives a relative measure of the *IOP*.

It should be borne in mind that the estimated *IOP* only provides a lower boundary for the *IOP* because the part attributed to disparity due to unobserved circumstances might be wrongly attributed to individual effort or error term.

Finally, Shapley decomposition was employed to decompose the total *IOP* into each family characteristic, which allows readers to understand the degree to which each family background contributes to the total *IOP*.

All statistical analyses were conducted with Stata 14.0 (Stata Corp., College Station, TX, USA).

## 3. Results

### 3.1. Secular Trend of Children’s Health Outcomes

The yearly descriptive statistics of participants by survey year are shown in Table 1. The upper panel of Table 1 presents the children’s characteristics. We found the average age changed slightly across the waves, and there was a slightly higher share of boys and more Han children and rural children in our sample. We also found that both the shares of calories drawn from protein and fat increased over time, indicating that the share of calories drawn from carbohydrates decreased steadily over time. The lower panel of Table 1 presents the various family environments of children. The average number of children per family in our sample was 1.74, which declined continuously between 1991 and 2006 but increased slightly in 2009 due to the relaxation of China’s family planning policy. A similar trend was also detected in household size. In addition, we observed an increasing household income per capita and improving sanitation conditions over time. Maximum education attainment and the BMI of parents also increased continuously over the years.

The growth status and weight status of children are shown in Table 2. Mean height and weight both followed an inverted-U trend and peaked in the year 2000. It should be noted that both indicators were strongly associated with the age structure of our sample in each year. The mean BMI increased with fluctuations. After standardization, we found an increasing trend in all three growth indicators. The share of overweight and obese children increased steadily over time, while that of underweight children still accounted for more than 20% of the total in 2011.

To present the secular trend of children’s physical growth and weight status in a more intuitive way, we also mapped the three normalized growth indicators and shares of three BMI categories in Figure 1. We found that all standardized growth indicators increased significantly, but standardized weight grew much faster than standardized height and resulted in a slowly growing BMI after standardization. The right lower panel of Figure 1 shows that the share of overweight and obese children increased quickly from 5.10% and 1.79% in 1991 to 11.21% and 8.60% in 2011, respectively (Figure 1). At the same time, the share of underweight children only declined slightly from 26.72% to 22.02%.

### 3.2. Association between Children’s Health Outcomes and Family Environments

We further adopted a multivariable linear regression model to investigate the association between three growth indicators and potential growth determinants. Similarly, a multivariable logistic model was employed to detect the association between the BMI groups and their risk factors. The results are presented in Table 3. The FE model was preferred due to the three model-specification tests (presented at the bottom of Table 3); the RE logit model was preferred as the panel-level variance component was significantly different from 0 (the LR test rejected the null hypothesis). We only present the results from the preferred models in Table 3 and show the results from other models in a Appendix A (Appendix A).

We found that individual heterogeneity (children’s age, gender, ethnicity, dietary structure, and residential region) had a significant impact on children’s growth status. For instance, boys and Han children had a higher risk of being overweight and obese. Children from urban areas were less likely to be underweight. More importantly, we found significant heterogeneity in children’s health status across different family environments even after adjusting for individual characteristics. In particular, children with fewer siblings tended to be heavier and have a higher BMI than children with more siblings. Moreover, they had a lower risk of being underweight but a higher risk of being overweight compared with their counterparts with more siblings. Interestingly, children who grew up in large families had a higher risk of being both underweight and overweight. The explanation of these results remains a topic for future research. Barring statistical (in)significance, household income had a positive impact on children’s physical growth and contributed to a higher risk of overweight. Sanitation conditions were also significantly associated with children’s physical growth and weight status. For instance, children who grew up in families with access to tap water had significantly greater standardized height, smaller standardized BMIs, and a higher risk of being underweight; children who grew up in families with an in-house toilet had significantly larger standardized BMIs and a higher prevalence of underweight and obesity; children who lived in a household with no near-house excreta removal had significantly lower standardized weight and standardized BMIs and a lower rate of underweight but higher risk of obesity. Children with well-educated parents tended to have a higher risk of being overweight, which was consistent with the positive association between education and weight status detected in lower-income countries [30]. In addition, a positive association trend was also detected between the parents’ maximum education level and children’s physical growth, but the coefficients were statistically insignificant. Finally, children whose parents had a higher BMI tended to be taller, heavier, and fatter; in addition, parents’ BMI scores were negatively associated with children’s risk of underweight but positively associated with children’s risk of overweight and obesity. Similar results were detected in other models (Appendix A). Barring statistical (in)significance, family environments were found to be associated with children’s physical growth and weight status.

To present the association between children’s health outcomes and various family environments in a more intuitive way, we also present the mean value of six health indicators by five family environment factors. The results are presented in Appendix A. Consistent with the results in Table 3, we found that children who grew up in small families, rich families, families with better sanitation conditions, and families in which the parents were well-educated and had a higher BMI tended to be taller, heavier, and fatter and have a lower rate of underweight but higher risk of overweight and obesity.

### 3.3. Disparity in Children’s Health Outcomes

The disparity in children’s physical growth and weight statuses is presented in Table 4. We found a significant increasing disparity in the children’s overweight and obesity risks, but no significant trend was detected in the disparity of children’s physical growth.

To determine how much of the disparity in children’s health outcomes could be attributed to the heterogeneity in family environments, we further decomposed the total variation in children’s health outcomes into six components, including individual heterogeneity and five family environments. The results are presented in Table 5 and Figure 2. Table 5 shows that more than 70% of the variation in standardized height, weight, and overweight could be attributed to family environments, and about 45% of the variation in standardized BMI was associated with family environments. About half of the disparity in underweight and obesity could be attributed to family environments.

We further decomposed the total disparity in children’s health outcomes in each wave and mapped the results in Figure 2. In general, the share of disparity in children’s physical growth associated with family environment tended to increase while that associated with individual characteristics (the yellow area) tended to decrease. Similar results were also observed in the disparity of underweight, overweight, and obesity. In addition, we found that the variations in BMI and the disparity in children’s obesity risk were mainly attributable to individual heterogeneity in the first six waves.

Finally, we decomposed the total disparity into between-group disparity and within-group disparity for the five family environmental factors and mapped the share of between-group disparity in each wave in Figure 3. We found that the between-group disparity was quite small compared with the within-group disparity. In particular, between-group disparity only accounted for about 1–6% of total disparity in the three standardized physical growth indicators and about 3–20% of total disparity in the three weight status indicators. We found a great variation in the share of between-group disparity across different waves, and a U-shaped trend was observed in most indicators and groups.

## 4. Discussion

Associated with rapid urbanization, economic growth, and cultural and technical changes, China has been transforming from a stage of receding famine to a stage of degenerative disease since the economic reform and opening-up policies in the late 1980s [1,6,31,32]. Consequently, a significant decline in under-nutrition was accompanied by increasing concern regarding obesity. On the one hand, the numbers of stunted and under-nourished children decreased significantly, but the nutrition and growth status of many vulnerable children has remained poor. For instance, left-behind children in remote rural areas still show a deficient intake of most nutrients [33], and motherless children are significantly shorter and lighter [34]. On the other hand, Chinese children are becoming taller and heavier, and the prevalence of overweight and obese has increased quickly since the 1980s [35,36,37].

In this study, we considered 20 years of data to detect the association between children’s growth status and five family environments and decomposed the disparity in children’s growth status into several components attributable to family environments. We found that the weight, height, and BMI of Chinese children increased steadily between 1991 and 2011. Consequently, the prevalence of overweight and obesity also experienced rapid growth during this period. These results were consistent with findings from previous studies [1,6,10,31]. The improving physical growth and increasing risk of overweight and obesity could be attributable to many factors, such as urbanization, nutrition transition, lifestyle change, and changing family environments. For instance, we found that urban children were less likely to be underweight. This may be due to the increasing availability of food—particularly, high-calorie food such as Western fast food—and decreasing opportunities for engagement in heavy physical activities such as household work and agricultural production [3,38,39].

More importantly, we found that the changing health statuses of children were significantly affected by various family environments. In particular, children from single-child families were found to be heavier and to have a higher BMI and, thus, less likely to be underweight but more likely to be overweight. Previous studies also found that children from single-child families were more likely to be over-fed and, thus, become significantly heavier and have a higher BMI than their counterparts with siblings, and the differences in health outcomes between children from single-child families and children with siblings could be attributable to higher consumption of animal-sourced food, higher frequency of eating Western fast food and drinking sweetened soft drinks, a higher share of meals eaten away from home, and more sedentary activity [10,40]. However, the easing of fertility control policies in recent years may slow the increasing prevalence of overweight and obesity and reduce the corresponding health burden for China [10]. Children who grew up in more affluent families tended to be taller and have a higher risk of being overweight. Previous literature has shown that income growth remains one of the major driving forces of nutrition transition in China [1,41], and children who grow up in rich families tend to consume more animal-sourced food and eat more frequently away from home [1]. Better sanitary conditions also had a significant impact on children’s growth. Previous literature has shown that inadequate sanitation might increase the risk of enteric infections and diarrhea, which further disrupts energy absorption, with a negative impact on children’s growth [42]. In addition, parents’ education attainment had a significant impact on children’s growth. Well-educated parents may have better nutrition knowledge and care more about children’s growth, and they have higher motivation to adopt a healthy lifestyle as role models for their children [17]. It should be noted that a father’s and mother’s education may have different impacts on children’s growth status. However, children who only live with one of their parents do not have observations of both parents, and we therefore only used the maximum education level of parents to capture the education background. Finally, children whose parents had a higher BMI were taller, heavier, and had a higher BMI, a lower risk of being underweight, and a higher risk of overweight and obesity. The intergenerational linkage of weight status between parents and children can be attributed to many factors, such as a peer effect from parents and recent genetic discoveries about the basis of human obesity [43,44].

In addition, we also found a huge disparity in children’s health outcomes. The disparities in health during childhood were particularly pernicious due to the health–poverty trap, which indicates that poor health in childhood could have a long-term negative impact on physical and cognitive development, education attainment, and future earnings [45,46,47]. The current literature claims that children’s health status is determined by demographic characteristics (e.g., age and sex), lifestyle (e.g., physical activity, sedentary behavior, smoking, and drinking), and circumstances such as the family environment [28,30,48]. In particular, the health disparity associated with the family environment is of special interest because these variables are considered a source of illegitimate inequalities [18,29]. Our decomposition showed that heterogeneity in family environments accounted for a major share of total disparity in standardized height, weight, and overweight and about half of the total disparity in standardized BMI, underweight, and obesity.

Finally, we found a U-shaped trend in the share of between-group disparity across the waves: between-group disparity decreased before 2000 and started to increase again in the early 2000s. In addition, the results showed that the share of disparity in all indicators attributable to the family environment tended to increase while those attributable to individual characteristics tended to decrease. The increasing disparity attributable to family environments calls for more precise policies to improve the nutrition and health status of children and reduce ex-ante inequality in children’s health outcomes. In poor rural areas, increasing food accessibility via the development of food markets [49,50,51], nutrition education for caregivers [52], and the promotion of biofortified crops [53] can reduce nutrition deficiency and improve the overall nutrition status of children living in poverty, which can further help these children catch up with their counterparts living in rich urbanized areas in terms of nutritional and health status. On the contrary, controlling the obesity epidemic should be at the core of nutrition policies in urbanized and economically well-developed areas, particularly in terms of building up a virtuous cycle in the family environment. For this reason, providing more nutritional education packages for pregnant women and part-time jobs for child-bearing women may have a long-term benefit for the whole family and even the wider society. Policies such as the taxation of food and beverages with added sugars and fats [3], emphasizing daily activity in school [3], labeling energy and nutrient contents for processed food and dishes served in restaurants [54,55], and promoting dietary knowledge in primary and middle schools to encourage healthy nutrition habits from an early age [10] can contribute to higher dietary quality and improve the health outcomes of children in China.

Our study contributes to the current literature in several ways: first, we considered 20 years of data to examine the secular trend of children’s growth status and its disparity; second, we investigated the impact of various family environments on children’s growth status; third, we estimated the share of disparity in children’s growth status associated with family environments by employing a decomposition method.

Several limitations of our study should be mentioned. First, previous literature found that one major contributor to the deteriorating health status in Chinese children was declining physical activity and growing levels of sedentary behaviors [3]. A recent study found that less than three-tenths of Chinese school-aged students met the guidelines for daily moderate-to-vigorous physical activity [56], and about four-tenths spent more than 2 h every day on screen-time viewing [57]. The overall physical fitness of children was 167% lower in 2014 compared with 1995, and obese boys experienced the largest decline in physical fitness indicators [3]. However, the CHNS only began to report physical activity data in 2004, and it would have significantly reduced the sample size and shortened the study period if we had included physical activity data in our sample. We thus assumed that the heterogeneity in physical activity was time-invariant, meaning that it could be controlled in the panel data model. Second, the growth curve of Chinese children is quite different from that of other countries, so the results should be interpreted with caution when used for international comparison. Third, this study was a pooled longitudinal analysis, which hampered causal references as well. Future studies employing more detailed data and appropriate statistical methods should be conducted to test the robustness of our findings.

## 5. Conclusions

This study found that the growth status (standardized height, weight, and BMI) of Chinese children improved steadily from 1991 to 2011. Consequently, the prevalence of overweight and obesity also increased significantly during this period. Meanwhile, the disparity in overweight and obesity has significantly increased over time. Further regression analysis found a significant association between family environments and children’s growth status and weight status. In addition, the disparity decomposition showed that about one-half to two-thirds of the disparity in children’s growth status and weight status could be attributable to heterogeneity in family environments, and the contribution of family environments to total disparity tended to increase over time.

## Figures and Tables

**Figure 1 children-08-00397-f001:**
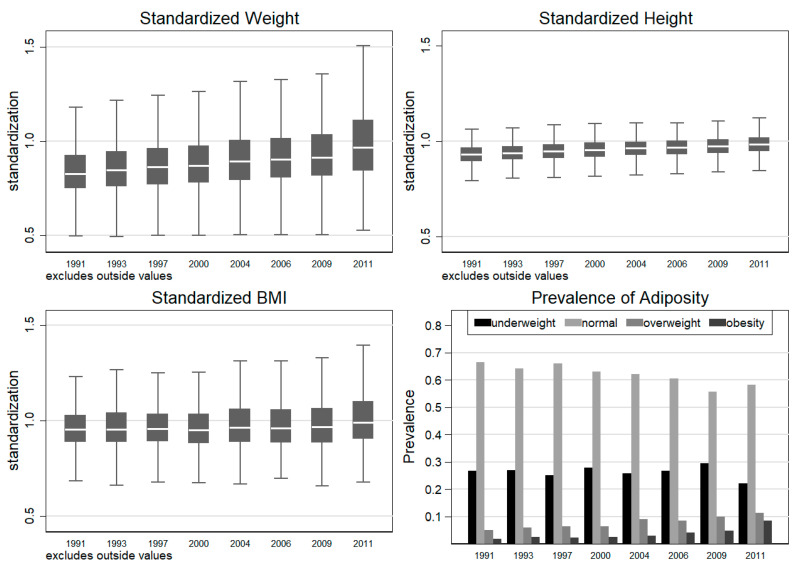
Secular trend of growth status.

**Figure 2 children-08-00397-f002:**
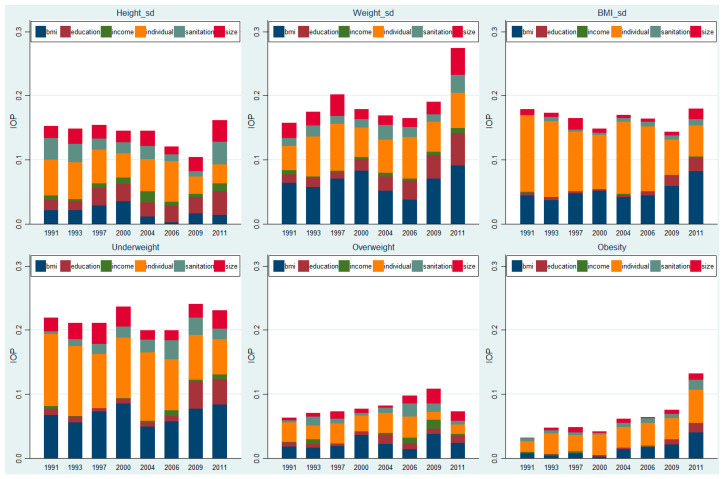
Secular trend of inequality of opportunity (IOP) in physical growth and weight status.

**Figure 3 children-08-00397-f003:**
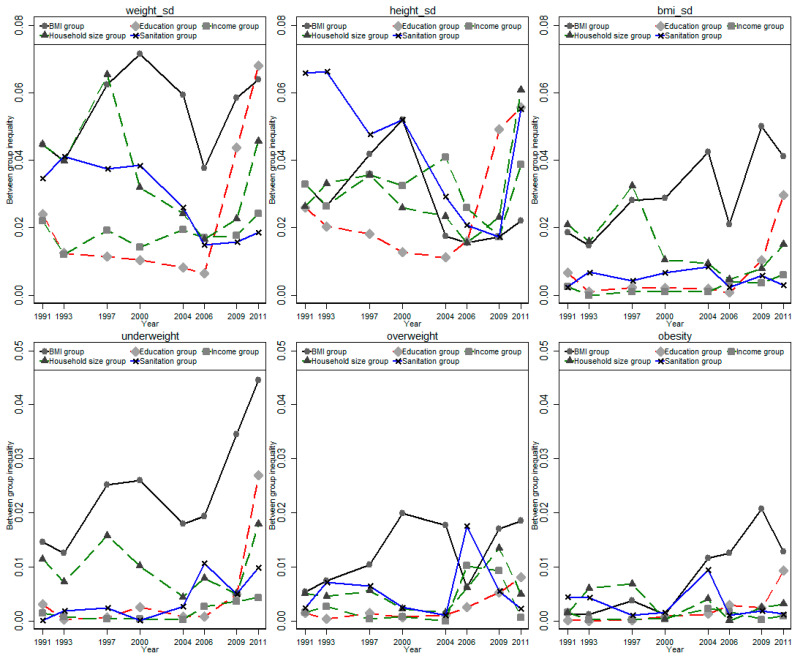
Secular trends of between-group disparity.

**Table 1 children-08-00397-t001:** Characteristics of participants from 1991 to 2011.

Year	1991	1993	1997	2000	2004	2006	2009	2011
Sample size	3080	2747	2405	2099	1440	1155	1066	1267
Individual factors								
Age	9.87 (5.37)	10.08 (5.08)	10.54 (4.78)	11.25 (4.68)	10.71 (5.21)	10.57 (5.15)	9.83 (4.93)	9.98 (4.72)
Girls	47.37% (0.50)	47.43% (0.50)	46.61% (0.50)	46.88% (0.50)	45.21% (0.50)	46.93% (0.50)	44.09% (0.50)	49.57% (0.50)
Han (ethnicity)	81.88% (0.39)	83.80% (0.37)	85.36% (0.35)	84.32% (0.36)	85.76% (0.35)	85.28% (0.35)	86.59% (0.34)	88.32% (0.32)
Regions (urban)	25.65% (0.44)	24.68% (0.43)	29.15% (0.45)	26.54% (0.44)	30.00% (0.46)	30.22% (0.46)	28.05% (0.45)	38.36% (0.49)
Dietary pattern								
Fat_share	21.97% (0.11)	22.04% (0.11)	25.24% (0.11)	27.41% (0.11)	28.01% (0.11)	29.53% (0.11)	30.79% (0.11)	33.49% (0.11)
Protein_share	11.73% (0.02)	12.04% (0.02)	12.07% (0.03)	12.16% (0.03)	12.52% (0.03)	12.45% (0.03)	12.89% (0.03)	14.05% (0.03)
Family environment								
Household scale								
Children	2.03 (0.91)	1.95 (0.84)	1.90 (0.84)	1.61 (0.72)	1.44 (0.62)	1.38 (0.58)	1.48 (0.64)	1.36 (0.59)
hhsize	4.77 (1.34)	4.71 (1.33)	4.49 (1.23)	4.37 (1.25)	4.25 (1.25)	4.36 (1.42)	4.71 (1.54)	4.35 (1.41)
ln(income)	7.60 (1.07)	7.65 (1.20)	7.96 (1.09)	8.17 (1.01)	8.34 (1.15)	8.45 (1.15)	8.81 (1.12)	9.05 (1.40)
Sanitation condition								
Tapwater	52.47% (0.50)	54.93% (0.50)	62.08% (0.49)	63.60% (0.48)	70.90% (0.45)	73.16% (0.44)	78.52% (0.41)	87.61% (0.33)
Toilet	12.92% (0.34)	17.69% (0.38)	25.82% (0.44)	32.11% (0.47)	40.49% (0.49)	45.28% (0.50)	50.47% (0.50)	70.24% (0.46)
No Excreta	42.89% (0.49)	45.58% (0.50)	56.76% (0.50)	59.70% (0.49)	69.79% (0.46)	71.60% (0.45)	74.48% (0.44)	81.22% (0.39)
Max_edu_parent	7.03 (3.69)	7.44 (3.59)	8.29 (3.31)	8.59 (3.27)	9.11 (2.99)	9.23 (3.12)	9.26 (2.99)	10.20 (3.31)
Max_BMI_parent	21.68 (2.23)	21.93 (2.22)	22.29 (2.27)	22.78 (2.59)	22.89 (2.64)	22.92 (2.63)	23.03 (2.87)	23.52 (3.17)

Note: Values in brackets are standard deviation; fat_share and protein_share refer to the share of energy drawn from fat and protein; children refers to the number of children; hhsize refers to household size; ln(income) is measured by household income per capita in the logarithm; tapwater indicates in-house or in-yard tap water; toilet indicates in-house flush or in-house toilet; no excreta refers to no near-house excreta removal; Max_edu_parent and Max_BMI_parent indicate the maximum education and BMI of parents.

**Table 2 children-08-00397-t002:** Growth status and weight status of participants from 1991 to 2011.

Year	1991	1993	1997	2000	2004	2006	2009	2011
Height (cm)	127.50 (28.08)	130.00 (26.32)	133.65 (25.43)	138.57 (25.33)	136.27 (29.38)	135.77 (28.28)	133.96 (27.25)	136.51 (25.59)
Weight (kg)	30.02 (15.51)	30.98 (15.07)	32.92 (14.91)	35.77 (15.28)	35.51 (16.98)	34.94 (16.81)	33.53 (16.63)	36.02 (17.37)
BMI (kg/m^2^)	17.06 (2.71)	17.12 (2.71)	17.33 (2.77)	17.61 (2.93)	17.80 (2.94)	17.68 (3.14)	17.40 (3.31)	18.19 (4.29)
Height_sd	0.84 (0.14)	0.86 (0.15)	0.88 (0.17)	0.89 (0.16)	0.91 (0.17)	0.92 (0.18)	0.95 (0.20)	1.01 (0.26)
Weight_sd	0.93 (0.05)	0.94 (0.05)	0.95 (0.05)	0.95 (0.05)	0.96 (0.06)	0.96 (0.06)	0.97 (0.06)	0.98 (0.06)
BMI_sd	0.97 (0.12)	0.97 (0.13)	0.98 (0.13)	0.97 (0.14)	0.99 (0.14)	0.99 (0.16)	0.99 (0.17)	1.04 (0.24)
Underweight	26.72% (0.44)	27.05% (0.44)	25.03% (0.43)	27.92% (0.45)	25.83% (0.44)	26.75% (0.44)	29.46% (0.46)	22.02% (0.41)
Overweight	5.1% (0.22)	6.08% (0.24)	6.53% (0.25)	6.57% (0.25)	9.10% (0.29)	8.48% (0.28)	9.85% (0.30)	11.21% (0.32)
Obesity	1.79% (0.13)	2.62% (0.16)	2.45% (0.15)	2.53% (0.16)	3.06% (0.17)	4.16% (0.20)	4.97% (0.22)	8.60% (0.28)

Note: Values in brackets are standard deviation; Height_sd, Weight_sd, and BMI_sd denote children’s height, weight, and BMI normalized by the WHO growth reference [20,21].

**Table 3 children-08-00397-t003:** Associations between family environments and children’s growth status and weight status.

	Height_sd	Weight_sd	BMI_sd	Underweight	Overweight	Obesity
Age	0.0030 ***	−0.0017 ***	−0.0080 ***	0.1405 ***	−0.0733 ***	−0.1850 ***
	(0.000)	(0.000)	(0.000)	(0.006)	(0.008)	(0.013)
Female				0.0279	−0.2543 ***	−0.6125 ***
				(0.059)	(0.081)	(0.119)
Han				−0.0478	0.2773 **	0.3175 *
				(0.081)	(0.125)	(0.183)
Fat_share	0.0068	0.0234	0.0122	0.4060	−0.0160	0.5098
	(0.005)	(0.015)	(0.015)	(0.260)	(0.370)	(0.529)
Protein_share	0.0565 ***	0.1878 ***	0.0822	−1.7720 *	3.9017 ***	6.2142 ***
	(0.019)	(0.062)	(0.062)	(1.065)	(1.402)	(1.983)
Urban				−0.3057 ***	0.0669	0.0128
				(0.072)	(0.095)	(0.136)
Sibling	−0.0011	−0.0094 ***	−0.0086 ***	0.2009 ***	−0.1359 **	0.0610
	(0.001)	(0.003)	(0.003)	(0.039)	(0.063)	(0.088)
hhsize	0.0005	0.0018	0.0015	−0.0530 **	−0.0937 ***	−0.0135
	(0.001)	(0.002)	(0.002)	(0.025)	(0.035)	(0.046)
ln(income)	0.0008 **	0.0021	0.0009	0.0085	0.0654 *	0.0063
	(0.000)	(0.001)	(0.001)	(0.023)	(0.036)	(0.049)
Tap water	0.0030 **	−0.0025	−0.0094 **	0.1505 **	−0.1100	−0.1452
	(0.001)	(0.004)	(0.004)	(0.062)	(0.095)	(0.138)
Toilet	−0.0015	0.0044	0.0086 *	0.1472 **	0.1168	0.2912 **
	(0.001)	(0.005)	(0.005)	(0.069)	(0.097)	(0.139)
Sanitation	−0.0003	−0.0054 *	−0.0055 *	−0.1286 **	0.0103	0.2738 **
	(0.001)	(0.003)	(0.003)	(0.057)	(0.089)	(0.131)
maxedu	0.0002	0.0007	0.0002	0.0131	0.0370 ***	0.0210
	(0.000)	(0.001)	(0.001)	(0.009)	(0.013)	(0.020)
BMI_parent	0.0006 **	0.0062 ***	0.0055 ***	−0.2383 ***	0.1766 ***	0.2401 ***
	(0.000)	(0.001)	(0.001)	(0.012)	(0.015)	(0.020)
Constant	0.8875 ***	0.7300 ***	0.9342 ***	2.2024 ***	−7.3634 ***	−9.5709 ***
	(0.008)	(0.026)	(0.026)	(0.349)	(0.500)	(0.717)
Observation	15,259	15,259	15,259	15,259	15,259	15,259
F/chi2	110.08 ***	7.48 ***	55.49 ***	819.59 ***	318.95 ***	330.88 ***
F test	3.23 ***	3.82 ***	2.22 ***			
LM test	2070.71 ***	2991.22 ***	831.44 ***			
Hausman test	304.38 ***	387.11 ***	115.22 ***			
LR test (rho = 0)				485.63 ***	110.80 ***	45.79 ***
Preferred model	FE	FE	FE	RE-logit	RE-logit	RE-logit

Note: Standard error in brackets. *, **, *** refer to significant levels at 10%, 5%, and 1%, respectively. Children’s characteristics include their age, gender, registered residence, ethnicity, and dietary pattern. Height_sd, weight_sd, and BMI_sd refer to various growth indicators normalized by the WHO growth reference [20,21].Maxedu refers to the highest education of parents; BMI_parent is the average BMI of parents. F test, LM test and Hausman test are three model specification tests, which are used to compare pooled ordinal least square estimation (OLS) and fixed effect model (FE), OLS model and random effect model (RE), and FE and RE, respectively. LR test is used to compare the logistic model and the random effect logistic model (RE-logit). Rho refers to the contribution of panel-level variance in total variance.

**Table 4 children-08-00397-t004:** Secular trend of disparity in health outcomes.

Indicator	1991	1993	1997	2000	2004	2006	2009	2011	Linear Trend
Height_sd	0.15412	0.14926	0.15535	0.14654	0.14714	0.12472	0.10627	0.16344	1.62
Weight_sd	0.15668	0.17343	0.19866	0.17781	0.16688	0.16477	0.18651	0.25882	−1.07
BMI_sd	0.17455	0.16935	0.16093	0.14348	0.16608	0.15783	0.13832	0.16362	−1.50
Underweight	0.21940	0.21075	0.21096	0.23606	0.19966	0.19957	0.24079	0.23086	0.67
Overweight	0.06315	0.07092	0.07292	0.07677	0.08185	0.09791	0.10799	0.07312	2.24 *
Obesity	0.03198	0.04737	0.04831	0.04150	0.06119	0.06380	0.07570	0.13185	3.44 **

Note: Three standardized growth indicators were measured by GE(0) (mean log deviation), and the three BMI groups were measured by DI (modified dissimilarity index). A linear trend test was conducted by the regression of the mean value in each year according to the time trend. *, and ** refer to *p* < 0.1, and *p* < 0.05, respectively.

**Table 5 children-08-00397-t005:** Inequality of opportunity.

Indicator	Height_sd	Weight_sd	BMI_sd	Underweight	Overweight	Obesity
Total variation	0.0875	0.2132	0.1229	0.2041	0.0696	0.0527
Bootstrap standard error	(0.000)	(0.000)	(0.000)	(0.005)	(0.010)	(0.016)
Individual characteristics	28.96%	25.48%	55.73%	45.78%	27.13%	52.15%
Household scale	14.31%	12.44%	5.22%	9.85%	13.47%	5.63%
Income	9.56%	5.36%	0.69%	1.25%	6.71%	2.53%
Tapwater, toilet, sanitation	17.07%	9.93%	1.60%	5.72%	8.46%	8.00%
Max_edu_parent	15.65%	11.39%	3.02%	4.27%	10.72%	6.34%
Max_BMI_parent	14.45%	35.41%	33.74%	33.12%	33.51%	25.30%

Note: Individual characteristics include age, gender, ethnicity, calorie sources, and residence region. Household scale includes the number of children and household size; Max_edu_parent and Max_BMI_parent indicate the maximum education and BMI of parents.

## Data Availability

Data will be provided upon request.

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
