# Peer review of "Growth and Weight Status in Chinese Children and Their Association with Family Environments"

_children, 2021, doi:10.3390/children8050397_

Round 1
Reviewer 1 Report
Thank you for the opportunity to review this manuscript, which investigates secular trends and disparity of growth status and body adiposity in Chinese children, as well as the impact of various family environments on children’s growth, from 1991 to 2011. Strengths include a large sample size, longitudinal data, and a robust study. Main weaknesses included English language errors and word choice.
General
1) Please review the manuscript for English language errors. Although overall, I understand what you’re communicating, there are many grammatical errors. For example, the first sentence in the first paragraph of the Introduction should be “An abundance of literature has shown …” The second sentence in the second paragraph of the Introduction should be “Since the early 1980, Chinese …” These are a few examples.
2) Relatedly, please check for verb tense usage and consistency. For example, please use past tense when referring to the work of previous researchers or old data (e.g., the last sentence in the first paragraph of the Introduction) and present tense when referring to a table in your manuscript (e.g., the first sentence in the first paragraph of the Results). Additional guidelines are available here: https://www.unl.edu/gradstudies/connections/writing-about-your-research-verb-tense.
3) Please be careful about word choice. As one example, “inequality” – in the field of health disparities and health equity – refers to differences in health between groups of people (e.g., race/ethnicity, income, sexual orientation) due to resource allocation, social disadvantage, etc. When I see “inequality” in your manuscript, my initial association is health of advantaged vs. disadvantaged groups, but it seems that your use of “inequality” is not that. Perhaps, “difference” or “differences” would be more appropriate.
Introduction
4) Related to word choice, in the Introduction, you write: “… individual characteristics are born with nature and unjustifiable …” Please be careful about saying this. What do you mean by “born with nature”? What do you mean by “unjustifiable”? When I read this, it implies that individual characteristics are innate and uncontrollable, whereas there are many individual characteristics (e.g., behaviors, attitudes, knowledge, etc.) that may be changed. You should specify that you mean biological/genetic individual characteristics.
5) Relatedly, to strengthen the distinction between individual vs. environmental factors, you could consider using a model (e.g., socio-ecological model) or referring to the “social determinants of health.” This would increase the study’s appeal to those in public health, sociology, etc.
Materials and Methods
6) In 2.2 Measurements of Variables, you mention the Chinese guidelines for BMI. Please reference these guidelines or add a citation, so that others outside of China will know the Chinese guidelines. Relatedly, you write that “results are not so different when the Chinese reference is employed …” I, and other readers, may be interested in seeing the differences, so please consider adding a supplemental table that compares the numbers using the Chinese guidelines vs. the WHO guidelines.
Results
7) Table 1 is very overwhelming and not easy to read. Perhaps, you can make the table landscape, use smaller font, or find a way to increase the spacing. You could also put the BMI-related data in a separate table. Relatedly, is the Note underneath related to Table 1 or part of the manuscript’s body? If a footnote, it should be smaller font.
Discussion
8) In the last sentences of the third paragraph of the Discussion, you write that children with higher risk of overweight/obesity had parents with higher BMIs, and attributed this to genetic discoveries. However, this relationship could also be related to modeling, parenting, etc. It might not just be genetic but behavioral and environmental, as well.
Author Response
Thank you for the opportunity to review this manuscript, which investigates secular trends and disparity of growth status and body adiposity in Chinese children, as well as the impact of various family environments on children’s growth, from 1991 to 2011. Strengths include a large sample size, longitudinal data, and a robust study. Main weaknesses included English language errors and word choice.
Responses: Thank you very much for your careful reading and valuable comments, which significantly increased the quality of the manuscript. We have tried our best to revise the manuscript under the comments provided by you and another reviewer. In addition, we also send our manuscript to MDPI English editing service to polish our writing. We hope the revised manuscript meet your criteria. The point-to-point responses are as follows:
General
1) Please review the manuscript for English language errors. Although overall, I understand what you’re communicating, there are many grammatical errors. For example, the first sentence in the first paragraph of the Introduction should be “An abundance of literature has shown …” The second sentence in the second paragraph of the Introduction should be “Since the early 1980, Chinese …” These are a few examples.
Responses: Thank you for your comments and sorry for our poor English. We have employed the MDPI English editing service to polish our manuscript. We hope the revised manuscript is easier to understand.
2) Relatedly, please check for verb tense usage and consistency. For example, please use past tense when referring to the work of previous researchers or old data (e.g., the last sentence in the first paragraph of the Introduction) and present tense when referring to a table in your manuscript (e.g., the first sentence in the first paragraph of the Results). Additional guidelines are available here: https://www.unl.edu/gradstudies/connections/writing-about-your-research-verb-tense.
Responses: Thank you for your suggestion. We carefully checked the whole manuscript and revised verb tense following the guidelines. Additionally, we also sent our manuscript to MDPI English editing service to polish our writing.
3) Please be careful about word choice. As one example, “inequality” – in the field of health disparities and health equity – refers to differences in health between groups of people (e.g., race/ethnicity, income, sexual orientation) due to resource allocation, social disadvantage, etc. When I see “inequality” in your manuscript, my initial association is health of advantaged vs. disadvantaged groups, but it seems that your use of “inequality” is not that. Perhaps, “difference” or “differences” would be more appropriate.
Responses: We agree that we should use inequality more carefully. We changed it to “disparity” and “variation” in most cases. But in some place “inequality” is still used because it is a terminology. For instance, inequality of opportunity (IOP) is a terminology widely used in social science.
Introduction
4) Related to word choice, in the Introduction, you write: “… individual characteristics are born with nature and unjustifiable …” Please be careful about saying this. What do you mean by “born with nature”? What do you mean by “unjustifiable”? When I read this, it implies that individual characteristics are innate and uncontrollable, whereas there are many individual characteristics (e.g., behaviors, attitudes, knowledge, etc.) that may be changed. You should specify that you mean biological/genetic individual characteristics.
Responses: Thank you for your comments. We cited these sentence from several literature and did not think about it carefully. We revised these citations as follows:
“However, individual characteristics are either given (e.g., gender) or under control of children and can be changed via children’s effort (e.g., dietary pattern, physical activity), while most family environments could be avoided by reasonable measures, and the health difference associated with family environments are preventable and unnecessary, which are usually considered as health disparities that denotes an unjust difference in children health.”
5) Relatedly, to strengthen the distinction between individual vs. environmental factors, you could consider using a model (e.g., socio-ecological model) or referring to the “social determinants of health.” This would increase the study’s appeal to those in public health, sociology, etc.
Responses: Thank you for your comments. We adopted the “social determinants of health” to distinguish individual vs. environmental factors in the manuscript.
Materials and Methods
6) In 2.2 Measurements of Variables, you mention the Chinese guidelines for BMI. Please reference these guidelines or add a citation, so that others outside of China will know the Chinese guidelines. Relatedly, you write that “results are not so different when the Chinese reference is employed …” I, and other readers, may be interested in seeing the differences, so please consider adding a supplemental table that compares the numbers using the Chinese guidelines vs. the WHO guidelines.
Responses: We added a table in the supplementary file (Table S1) to compare the difference between Chinese guidelines and the WHO guidelines. We did not present all results estimated using Chinese guidelines, because it will add too many tables and figures. However, data and code will be available upon request. So that readers can conduct all empirical analyses by themselves to test the replicability of our findings.
Results
7) Table 1 is very overwhelming and not easy to read. Perhaps, you can make the table landscape, use smaller font, or find a way to increase the spacing. You could also put the BMI-related data in a separate table. Relatedly, is the Note underneath related to Table 1 or part of the manuscript’s body? If a footnote, it should be smaller font.
Responses: Thank you for your suggestion. We separated growth status of participants from Table 1 and presented this information in Table 2. We also reduced the font of number and word in Table 1 to make it more readable.
Note underneath Table 1 is related to Table 1. We have reduced the font of these words.
Discussion
8) In the last sentences of the third paragraph of the Discussion, you write that children with higher risk of overweight/obesity had parents with higher BMIs, and attributed this to genetic discoveries. However, this relationship could also be related to modeling, parenting, etc. It might not just be genetic but behavioral and environmental, as well.
Responses: Thank you for your comments. We agree that the intergenerational linkage of health could be attributable to many factors in addition to genetic. We revised this sentence as follows:
The intergenerational linkage of weight status between parents and children could be attributable to many factors such as peer effect from parents, and recent genetic discoveries about the basis of human obesity.

Reviewer 2 Report
This is a very interesting study on the secular trends of physical growth and weight status among Chinese children within a 20 years period and the association of those variables with family characteristics. The decomposition analysis of the discrepancies in physical growth and health indicators is very interesting. The methodology and the results are clearly presented and the conclusions are in line with the results.
I am concerned about the use of body adiposity in the title and several points of the manuscript. Usually, referring to body adiposity means that there are some adiposity measurements. Of course BMI shows a strong correlation with body adiposity, but I do not believe it should be used as a measure of body adiposity. In my opinion the manuscript refers more to weight status than to body adiposity.
In addition, I believe that the statement “a U-shape trend was observed in most indicators and groups” needs a more specific explanation in the text.
Finally, a careful english editing would improve the manuscript.
Author Response
This is a very interesting study on the secular trends of physical growth and weight status among Chinese children within a 20 years period and the association of those variables with family characteristics. The decomposition analysis of the discrepancies in physical growth and health indicators is very interesting. The methodology and the results are clearly presented and the conclusions are in line with the results.
Responses: Thank you very much for your confirmative review report. We are highly encouraged to revise the manuscript under the comments provided by you and another reviewer. Additionly, we also send our manuscript to MDPI English editing service to polish our writing. The point-to-point responses are as follows:
I am concerned about the use of body adiposity in the title and several points of the manuscript. Usually, referring to body adiposity means that there are some adiposity measurements. Of course BMI shows a strong correlation with body adiposity, but I do not believe it should be used as a measure of body adiposity. In my opinion the manuscript refers more to weight status than to body adiposity.
Responses: Thank you for your suggestion. We change all “body adiposity” to “weight status”.
In addition, I believe that the statement “a U-shape trend was observed in most indicators and groups” needs a more specific explanation in the text.
Responses: We add more discussion about the “U-shape trend” to clarify the definition in the text. We hope now the statement is clear and easy to understand.
Finally, a careful english editing would improve the manuscript.
Responses: We employed the MDPI English editing service to polish our writing. We hope the new manuscript is easy to understand.
